# A tool to check whether a symmetry-compensated collinear magnetic material is antiferro- or altermagnetic

Andriy Smolyanyuk[1], Libor Šmejkal[2] and Igor I. Mazin[3]*

**1** Institute of Solid State Physics, TU Wien, 1040 Vienna, Austria
**2** Johannes Gutenberg Universität Mainz, Mainz, Germany
**3** George Mason University, Fairfax, USA

*imazin2@gmu.edu

May 9, 2024

## Abstract

Altermagnets (AM) is a recently discovered class of collinear magnets that share some properties (anomalous transport, etc) with ferromagnets, some (zero net magnetization) with antiferromagnets, while also exhibiting unique properties (spin-splitting of electronic bands and resulting spin-splitter current). Since the moment compensation in AM is driven by symmetry, it must be possible to identify them by analyzing the crystal structure directly, without computing the electronic structure. Given the significant potential of AM for spintronics, it is very useful to have a tool for such an analysis. This work presents an open-access code implementing such a direct check.

## 1 Introduction

Magnets have recently been classified according to symmetry transformations into three types: ferromagnets (FM), antiferromagnets (AFM), and altermagnets (AM) [1–3]. Ferromagnets or

ferrimagnets (including Luttinger-compensated ferrimagnets, see Ref. [2]) exhibit net magnetization, which breaks the time-reversal symmetry in the electronic structure. On the other hand, antiferromagnets exhibit opposite spin sublattices coupled by translation and/or inversion, symmetry transformations that lead to time-reversal symmetric energy bands and zero magnetization. On the contrary, in *altermagnets* the opposite spin sublattices are coupled by symmetry operations (as in AFM, but not in FM) that are not inversions or translations, leading to an unusual combination of time-reversal symmetry broken electronic structure with alternating *in both coordinate and momentum spaces* spin polarization and zero net magnetization [1, 4, 5], and spin-split electrons bands (as in FM, but not in AFM). The spin split bands break time-reversal symmetry as in FM, but not in AFM. Additionally, the alternating spin splitting follows a d-, g-, or i-wave symmetry which is distinct from the symmetry of spin splitting in FM.

This alternating in the momentum space spin polarization can be expanded in spherical harmonics (pretty much as it is being done in the theory of unconventional superconductivity), and, depending on the underlying symmetry, can exhibit d-, g-, or i-wave magnetization density with 2, 4 or 6 spin-degenerate nodal surfaces [1].

Remarkably, many unusual effects related to altermagnetism were predicted. They include anomalous Hall effect [4, 6], crystal magnetooptical Kerr effect [5, 7, 8], large nonrelativistic spin splitting [4, 9, 10], spin-polarised longitudinal and transverse currents [11–13], giant and tunneling magnetoresistance [13, 14], nonlinear Hall effect [15], topological phase transitions [16], finite momentum Cooper pairing [17], anisotropic Andreev reflection [18], unconventional Josephson effect [19], magnon spin splitting [20], ferroically ordered multipoles [16, 21], nonunitary triplet superconductivity with the order parameter that averages to unitary and suppressed Andreev reflection at an interface with a conventional superconductor [22]. Altermagnetism can be found in diverse material families important for studying its application in spintronics, correlated states of matter, superconductivity, or semiconductor electronics (see also comprehensive list of references in perspective article [3]). Quadratic momentum-dependent spin splitting was also experimentally confirmed in photoemission spectra in MnTe altermagnet [23]. As of now, several candidate materials have been identified, but in each case, it was effected by manually checking the symmetry operations and/or calculating the band structure and verifying that it is spin-split. Furthermore, since the latter test cannot distinguish between AM and compensated FM, there is considerable confusion in this regard [24].

This work aims to create a program (and a library) which takes a crystal structure and a magnetic pattern of the material and decides whether it is antiferromagnetic or altermagnetic (it is trivial to rule out ferromagnetic materials). The input requested from the user is information about a crystal structure (various crystal structure formats are supported) and the magnetic pattern. Note that a further classification of a given altermagnet into one of ten classes, as suggested in Ref. 1, is out of the scope of this work.

## 2 Methodology

### 2.1 Symmetry and magnetism: bipartite lattice

General symmetry conditions have been deduced that distinguish AM from AFM [3, 4]. Here we provide a simple tool that would allow, by just looking at crystal structure and magnetic pattern, to say with certainty whether a material (existing or hypothetical) is AM. In the following, we will describe the ingredients for such a tool.

Let us assume that in the nonmagnetic parent compound the two sublattices that will be

assigned the "up" and "down" spins (we will call them, in short, the up and down sublattices) are related by a spatial symmetry operation $\hat{O}$, ignoring for a moment a possibility to have multiple species of atoms and focus only on magnetic atoms. Note that unless such an operation exists, the net magnetization is not required to be zero by symmetry (albeit it may be strictly zero by virtue of the Luttinger theorem [2]), and thus the material in question is, by symmetry, a ferrimagnet, and not subject to this study. Thus, we further assume spin compensation in the system, since ruling out that the system is ferromagnetic is trivial: the issue is to distinguish between AM and AFM states.

At first, it is instructive to examine what symmetries are responsible for up and down bands to be degenerate, i.e., when we are dealing with AFM material. By definition, $\hat{O}\hat{F}E_{\mathbf{k}\uparrow} = E_{\hat{O}\mathbf{k}\downarrow}$. Here, $\hat{O}$ is a spatial symmetry operation, and $\hat{F}$ flips the spins. We assume that space and spin coordinates are decoupled and that spin is a pseudoscalar quantity, i.e., merely "up" or "down" and not a pseudovector (which is commonly used when the structure is described by magnetic space group formalism [25]).

If $\hat{O}$ is an integer lattice translation, $\hat{O} = \hat{\mathbf{t}}$, then $\hat{O}E_{\mathbf{k}\uparrow} = E_{\mathbf{k}\uparrow}$, and therefore $E_{\mathbf{k}\uparrow} = E_{\mathbf{k}\downarrow}$, due to a spin flip operation mapping up and down sublattices to each other.

If this operation is a pure inversion, then the material is centrosymmetric, and by the action of inversion $E_{\mathbf{k}\uparrow} = E_{-\mathbf{k}\uparrow}$ and $E_{\mathbf{k}\downarrow} = E_{-\mathbf{k}\downarrow}$. Since the inversion mapping the up-sublattice onto the down-sublattice is followed by a spin flip, $E_{\mathbf{k}\uparrow} = E_{-\mathbf{k}\downarrow}$. Combining these expressions proves that $E_{\mathbf{k}\uparrow} = E_{\mathbf{k}\downarrow}$. Thus, in these two cases, spin-up and spin-down bands are degenerate and the material is AFM. Note that in this case an inversion center is located at the midpoint between spin-up and spin-down sites.

As we see, if the net magnetization is strictly zero, it may be not dictated by symmetry, which is the case of a Luttinger-compensated ferrimagnet, or result from one or more crystal symmetry operation mapping one spin sublattice upon the other. This symmetry operation may be (a) a lattice translation, (b) spatial inversion, or (c) another operation that is neither translation nor inversion. If all symmetry operations performing such matting belong to group (c), the material is altermagnetic and its bands are spin-split at general k-points. Thus, to check if the material is AM, one needs to confirm that there are no translations and no inversions that map one spin sublattice onto the other while confirming that there is some other symmetry responsible for such mapping.

## 2.2 AFM vs AM: examples

The idea can be illustrated using NiAs structure prototype as an example (see Fig. 1a). This structure contains an inversion symmetry which has different action depending on the atom under consideration: two As atoms are mapped onto each other, but Ni is left untouched by the inversion, since Ni atom is mapped to the equivalent (by translation) one in the next unit cell (the action of inversion is illustrated by various dashed lines coming from the center in Fig. 1a). If As is substituted by Fe and Ni by O, as in high-pressure FeO compound [27], the material is AFM and inversion followed by spin-flip maps Fe-up onto Fe-down, as illustrated in Fig. 1b. However, if As is substituted by Te and Ni by Mn as in MnTe [28] (see Fig. 1c), inversion followed by a spin-flip no longer maps up sublattice onto down one, and the material is AM. Note that only one unique magnetic pattern is possible in both cases when the primitive unit cell is considered.

In the previous example, the position of magnetic atoms favoring AFM-interaction was responsible for the transition between AFM or AM state. Another mechanism of AFM-AM transition can be illustrated using LiMnPO$_4$ structure (see Fig. 2, where Mn atoms are colored red and blue; other atoms, nonmagnetic, are gray). For this material AFM order is reported [29,30] (Fig. 2a): in this case, inversion followed by spin-flip will map up and down sublattices onto each other. If it would be possible to swap spins of atoms on one of the body diagonals (as

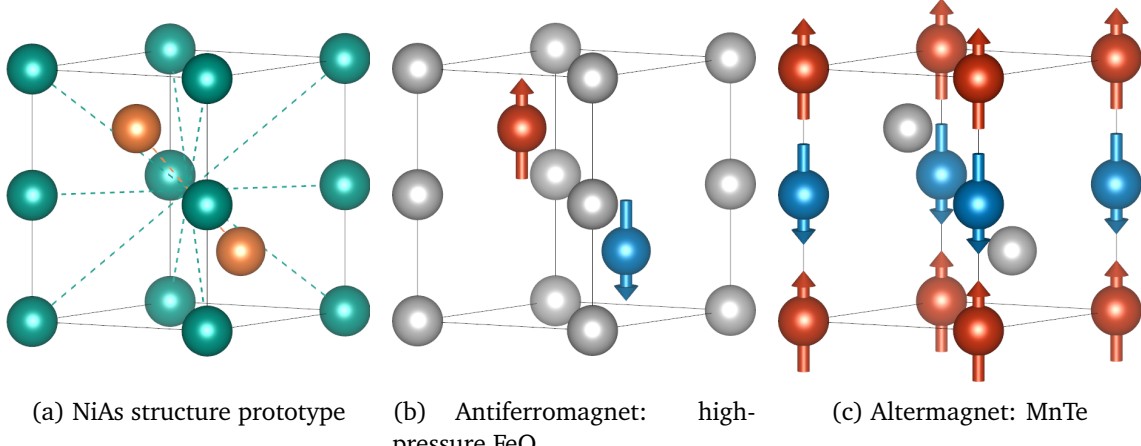

(a) NiAs structure prototype     (b) Antiferromagnet: high-pressure FeO     (c) Altermagnet: MnTe

Figure 1: Depending on what position a magnetic atom in (a) NiAs structure prototype occupies, it is possible to obtain (b) AFM , as in FeO, or (c) AM material, as in MnTe compound. Dashed lines connect atoms related by spatial inversion. Teal and orange spheres are Ni and As atoms; red and blue spheres represent spin up and down sublattices occupied by either Fe or Mn; non-magnetic atoms are gray. For the sake of illustration, the information about the spin is also denoted by arrows. Here and in other plots, VESTA software [26] was used to visualize crystal structure.

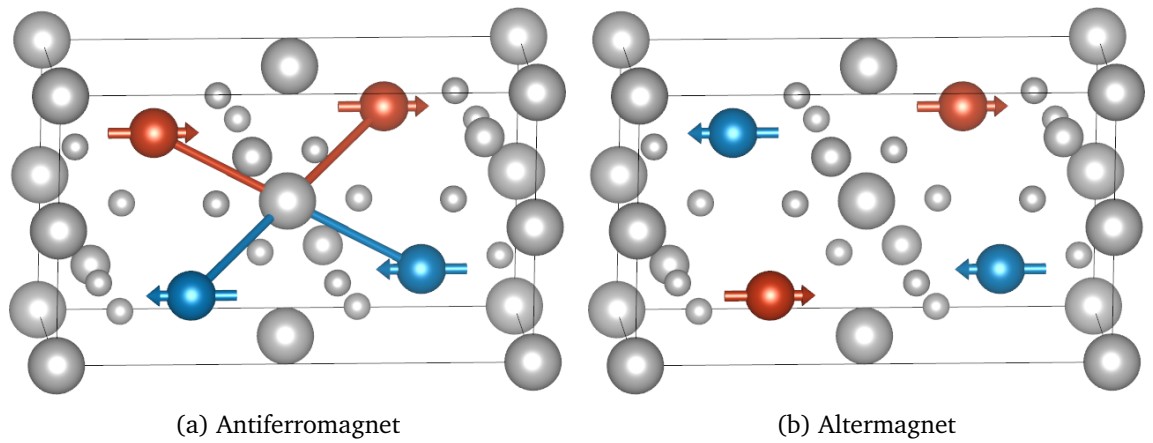

(a) Antiferromagnet     (b) Altermagnet

Figure 2: An illustration of AFM-AM transition driven by the type of magnetic pattern, for which $LiMnPO_4$-structure is used. Mn atoms are red and blue for spin up and down sublattices; non-magnetic atoms are gray.

illustrated in Fig. 2b), up and down sublattices can no longer be related by such an operation, and the material would be AM.

## 2.3 Structures with multiple magnetic atoms

When multiple magnetic ions are present in the crystal structure, some extra considerations are necessary.

Depending on how they behave under the action of symmetry operations, different points in a crystal can be assigned different labels called Wyckoff labels or Wyckoff positions [31], forming a disjoint set. For example, if only a mirror plane symmetry is present, points on this plane will behave differently under this symmetry than other points, thus obtaining a different label.

Each Wyckoff position has a multiplicity: the number of points that are equivalent by symmetry. A set of such points is called a Wyckoff orbit. If a given material consists of multiple atomic species, the up and down sublattices will be partitioned into disjoint sets by species. For each species, atoms will be further partitioned into symmetry-related groups (orbits). Given that different Wyckoff orbits are not symmetry-related, our task is to check each orbit separately for the possibility of generating spin splitting. Thus, for a material to be AM, we need to find at least one Wyckoff orbit, for which up and down sublattices are not related by a spin-flip followed by inversion or/and translation.

Again, we assume that each Wyckoff orbit that contains magnetic atoms has zero net magnetic moment. If that is not the case, we are dealing either with ferro- or ferrimagnet.

To determine if the state is AM or AFM, Wyckoff orbits with non-magnetic atoms can be neglected since they preserve spin flip symmetry operation. However, information about them is still needed to determine the symmetry of the parent non-magnetic structure.

## 2.4 Treatment of non-primitive unit cells

For the symmetry analysis of the given structure, we are using `spglib` library [32,33]. However, for non-primitive cells `spglib` restricts the search of symmetry operators by the point group of the lattice formed by the basis vectors of the non-primitive unit cell [34]. This is inconsistent with a usual approach in crystallography, where the search for symmetry operators is restricted by the point group of the primitive lattice and the approach we employ. Thus, if the non-primitive cell is analyzed, `spglib` will provide fewer symmetry operators than expected.

In this case, we will obtain the symmetry operators for the primitive cell instead and augment them with appropriate fractional translations that should appear when the corresponding non-primitive cell is considered and select the ones that are consistent with a given magnetic pattern. For example, if $P$ is a point group of a primitive cell and the non-primitive cell of interest is the 2x1x1 supercell, the resulting group is $P + \{E, (\frac{1}{2}, 0, 0)\}P$, where $E$ is an identity and $\{\hat{O}, \vec{t}\}$ denotes Seitz symbol, i.e., the amount of symmetry operations is doubled.

The procedure to obtain fractional translations is the following. The non-primitive cell (supercell) $S$ can be constructed from a given primitive cell $P$ with the help of an integer-valued transformation matrix $T$:

$$S = TP. \tag{1}$$

The ratio of volumes of the supercell and primitive cell is an integer given by the determinant of $T$, $n = \det(T)$. All possible supercells can be grouped based on the value of $n$. Although, in general, the amount of possible unit cells for a supercell with a fixed $n$ is infinite, they can be separated into a finite amount of classes based on the structure of Hermite Normal Form (HNF) $H$ representation of the matrix $T$ [35]. For example, for a parent square lattice, there are three classes of supercells with $n = 2$: as illustrated in Fig. 3.

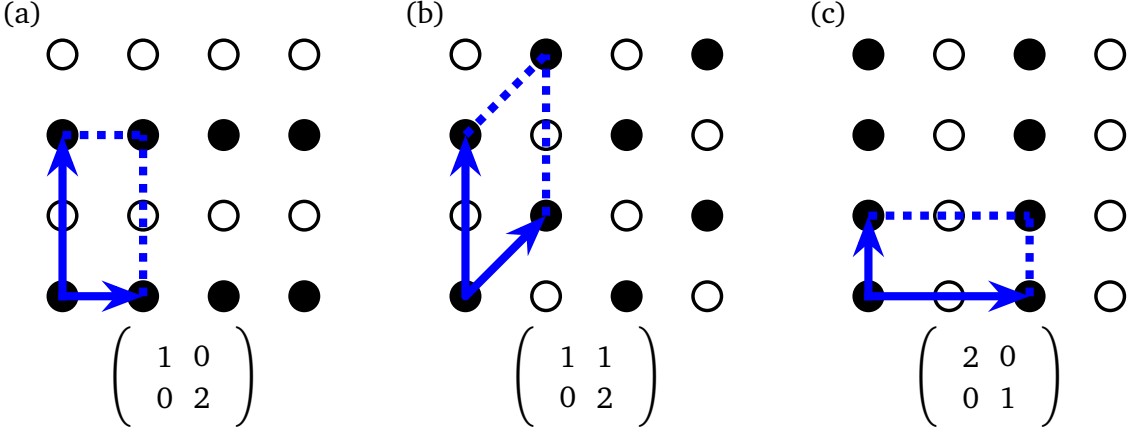

Figure 3: Three distinct classes of $n = 2$ supercells for a parent square lattice and respective HNFs: (a) horizontal stripes, (b) checkerboard pattern, and (c) vertical stripes. Equivalence by symmetry is not taken into account: if accounted for 90° rotation, supercells in (a) and (c) are the same.

Typically, one would be interested in labeling/coloring the atoms in each obtained supercell to enumerate all possible alloys/magnetic patterns that can be derived from the parent primitive cell. However, if such labeling is not performed, all supercells within the same $n$ will be equivalent since no symmetry breaking is introduced, the fact that we will exploit later.

We want to obtain the translation operations introduced by the transformation $T$ that transforms a primitive cell $P$ to a non-primitive cell $S$. Now, the cell $S$, the corresponding HNF $H$, and the supercell formed by the diagonal of $H$ are equivalent (since no symmetry breaking was introduced and non-magnetic structure is considered at this point). The latter tells us how many times in what direction the cell $P$ was multiplied. If we assume that the diagonal of $H$ is $(m, n, l)$, the fractional translations of interest in the basis of $H$ are $(\frac{i}{m}, \frac{j}{n}, \frac{k}{l})$ where $i \in [0, m-1]$, $j \in [0, n-1]$ and $k \in [0, l-1]$. After enumerating all translations, what is left to do is to transform them into the basis of $S$. The magnetic pattern is introduced after the cell $S$ and corresponding fractional translations are defined. For example, in Fig. 4 the algorithm is illustrated for the $\sqrt{2} \times \sqrt{2}$ supercell of the square lattice.

There are some rare cases of magnetic materials where the magnetic unit cell is larger than the chemical one, but the number of symmetry operations is smaller than the amount obtained in the previously described procedure [36]. In such a case, the obtained augmented list of symmetry operations should be reduced to be consistent with a given magnetic pattern. An example of such a material, which is altermagnetic, is $MnSe_2$ and given in Fig. 5 (with a magnetic pattern as reported in [37], denoted as $MnSe_2$-I; for more examples see Ref. [36]). Non-magnetic $MnSe_2$ has 24 symmetry operations, while the magnetic compound has the magnetic unit cell with three times the volume of the chemical unit cell but contains only 8 symmetry operations (as illustrated in Fig. 5b).

## 2.5 Action of symmetry operations

A practical way to check the action of symmetry operation on atomic position $\vec{r}$ is to represent proper and improper rotations as a $3 \times 3$ matrix $\hat{R}$ and a fractional translation as a 3-elements vector $\vec{\tau}$. Then the position $\vec{r}$ is left invariant by a symmetry operation if

$$\hat{R} \cdot \vec{r} + \vec{\tau} = \vec{r} + \vec{A}, \tag{2}$$

where $\vec{A} = m\vec{a}_1 + n\vec{a}_2 + k\vec{a}_3$ with $\vec{a}_i$ being lattice vectors and $m$, $n$, $k$ integers, meaning that all points related by lattice translations are considered to be equivalent. If all operations are

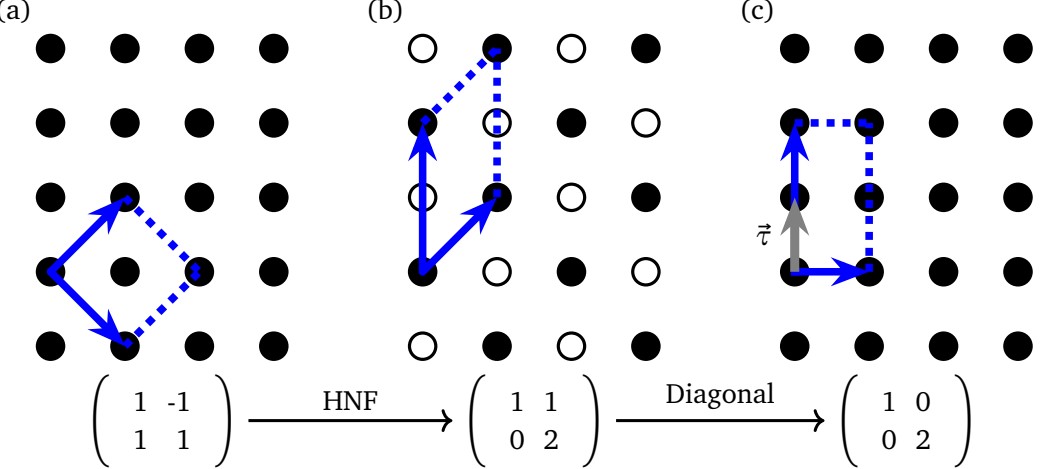

Figure 4: An illustration of how to obtain the fractional translations for a $\sqrt{2} \times \sqrt{2}$ supercell of the square lattice: (a) a $\sqrt{2} \times \sqrt{2}$ unit cell (blue dotted lines) and the corresponding basis vectors (blue arrows), (b) the HNF of the $\sqrt{2} \times \sqrt{2}$ unit cell, which gives the unit cell class representative: if bipartite lattice is formed, checkboard pattern would be realized; (c) the diagonal matrix obtained from the HNF gives a new unit cell that highlights the necessary fractional translations (gray vector $\vec{\tau} = (0,1)$).

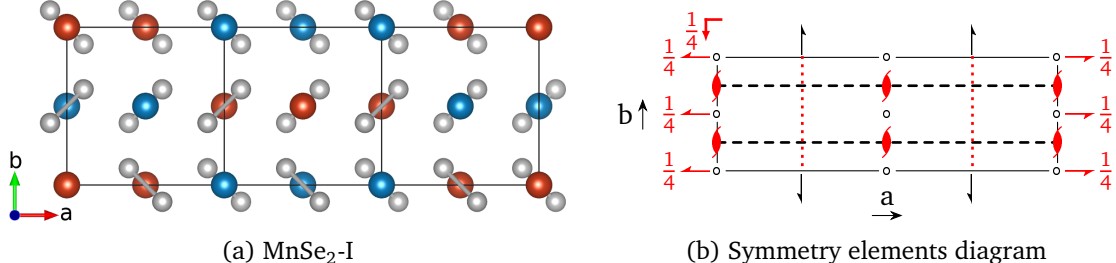

(a) MnSe$_2$-I                                         (b) Symmetry elements diagram

Figure 5: (a) Crystal structure and magnetic pattern of MnSe$_2$-I altermagnet: Mn atoms are red and blue for spin up and spin down sublattices; non-magnetic Se atoms are gray. The size of the magnetic unit cell is three times the volume of the chemical unit cell; the chemical unit cell is highlighted as well. (b) Symmetry elements that are present in this system; the spatial symmetries followed by a spin flip are red. For the meaning of symbols see Chapter 1.4 of Ref. 31.

expressed in the fractional coordinate system, the right hand side of Eq. (2) can be substituted by a vectorized module 1 operation:

$$\hat{R} \cdot \vec{r} + \vec{\tau} = \mathrm{mod}(\vec{r}, 1), \tag{3}$$

which conveniently accounts for translational equivalence.

Let us illustrate the use of Eq. (2). By definition of inversion, $\hat{I}\vec{r} = -\vec{r}$. Here, inversion can be represented by an inversion center at the origin, i.e., taking the negative of $3 \times 3$ unity matrix for $\hat{I}$ with no fractional translation (i.e., $\vec{\tau} = \vec{0}$). Thus, from Eq. (2): $\hat{I}\vec{r} = \vec{r} + \vec{A}$. Combining these two expressions one arrives at $\vec{r} = \frac{1}{2}\vec{A}$, which gives 8 points invariant by inversion: $\vec{r} = a_1\vec{a}_1 + a_2\vec{a}_2 + a_3\vec{a}_3$ with $a_i$ being equal to either 0 or $\frac{1}{2}$.

# 3   Program usage

In this manuscript, we describe a program that allows to determine if the material of interest is AM or AFM by providing the corresponding structure file and magnetic pattern. To handle multiple structure file formats `ase` library [38,39] is employed; thus, such formats as CIF [40] and POSCAR [41] are supported among the others.

Following the ideas from the section above, the algorithm is the following: if every magnetic atom with spin-up can be mapped on a spin-down atom using either inversion or translation followed by a spin flip, then the system is antiferromagnetic; thus, for the system to be AM, there should be at least one spin-up-down pair, atoms of which cannot be mapped by those symmetry operations.

Given that the set of all atoms in the structure naturally splits into symmetry-related groups (Wyckoff orbits), one needs to perform the analysis mentioned above for each Wyckoff orbit separately. If magnetic atoms in all Wyckoff orbits can be mapped by inversion or translation followed by a spin flip, then the material is antiferromagnetic. Otherwise, it is an altermagnet. The analysis of the symmetry of the given structure and the determination of Wyckoff orbits is done using `spglib` library [32,33].

The program can be installed from PyPi:

```
$ pip install amcheck
```

The source code is located in `github` repository and is accessible at the link: https://github.com/amchecker/amcheck.

Given the structure information stored in one of the supported formats, the program can be executed via the following command:

```
$ amcheck structure_file
```

and the information on how to run it can be requested using:

```
$ amcheck --help
```

giving the output specified in Listing 1.

`amcheck` will loop over all Wyckoff orbits, and the user will be asked to type spin designation for each atom in orbit using "u"/"U" symbol for spin-up, "d"/"D" for spin-down or "n"/"N" to mark the atom as non-magnetic (the entire orbit can be marked as non-magnetic using a single "nn"/"NN" designation). To help with this process, an auxiliary structure file in POSCAR format [41] will be created, and the user can open it with crystal structure visualization software. After all input data is provided, the analysis result will be printed in the last line.

An example of running an interactive session with the program is provided in the Listing 2. Note that it is possible to specify multiple structure files: the program will internally loop over

them. If a non-primitive cell is given, the user will be asked to use it as is or to analyze the corresponding primitive cell instead.

If the automatized search is of interest, one can use `amcheck` package as a library, and the code snippet of the typical usage is given in Listing 3.

Once the material is determined as altermagnetic, one could be interested in its transport properties and determine the allowed by symmetry form of the conductivity tensor, which in static limit describes the Anomalous Hall Effect (AHE) and in dynamic limit (X-ray) magnetic circular dichroism (X)MCD. This type of analysis can be performed using the following command (see an example in Listing 4):

```
$ amcheck --ahc structure_file
```

The user will be asked to enter the local magnetization vector for each atom, and the determination of the corresponding magnetic space group will be performed by `spglib`. Thus, the effect of spin-lattice coupling due to spin-orbit interaction is taken into account. Once the symmetry and antisymmetry operations are determined, the form of the conductivity tensor is obtained by symmetrizing a "seed" matrix $S$ following the tensor transformation rules described in Ref. 42:

$$\sigma = \sum_{R_i} R_i^{-1} S R_i + \sum_{R'_i} R'^{-1}_i S^T R'_i, \tag{4}$$

where $R_i$ are symmetry elements and $R'_i$ are antisymmetry elements.

Note that the obtained conductivity tensor is defined w.r.t cartesian coordinates as defined by a corresponding input structure file, which might not necessarily be aligned with crystallographic axes. So, a comparison with the experiment or reported tables should be made carefully. This convention is employed to aid the comparison with computational results since not all codes enforce the convention employed by crystallographers.

# 4 Conclusion

There is a significant interest of the scientific community in a novel magnetic phase called altermagnetism. However, the amount of suggested altermagnets is scarce, and a tool that would boost the discovery of new candidates would be of great use.

In this paper, we present an open-access code that checks if the material is altermagnetic or antiferromagnetic based on the symmetries obeyed by its crystal and magnetic structure.

**Funding information**   A.S. was supported by the Austrian Science Fund (FWF) through the project P33571 "BandITT". L.S. acknowledges funding from Deutsche Forschungsgemeinschaft (DFG) grant no. TRR 288 - 422213477 (project A09) and support from the Johannes Gutenberg-Universität Mainz TopDyn initiative (project Alterseed). I.I.M. was supported by the Army Research Office through grant # W911NF2220173.

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

```
usage: amcheck [-h] [--version] [-v] [-s SYMPREC] [-ms MAG_SYMPREC]
               [-t TOL] [--ahc]
               file [file ...]

A tool to check if a given material is an altermagnet.

positional arguments:
  file                  name of the structure file to analyze

optional arguments:
  -h, --help            show this help message and exit
  --version             show program's version number and exit
  -v, --verbose         verbosely list the information during the
                        execution
  -s SYMPREC, --symprec SYMPREC
                        tolerance spglib uses during the symmetry
                        analysis
  -ms MAG_SYMPREC, --mag_symprec MAG_SYMPREC
                        tolerance for magnetic moments spglib uses
                        during the magnetic symmetry analysis
  -t TOL, --tol TOL, --tolerance TOL
                        tolerance for internal numerical checks
  --ahc                 determine the possible form of Anomalous Hall
                        Coefficient
```

Listing 1: `amcheck` help output.

```
$ amcheck FeO.vasp MnTe.vasp
===========================================================
Processing: FeO.vasp
-----------------------------------------------------------
Spacegroup: P6_3/mmc (194)
Writing the used structure to auxiliary file:
check FeO.vasp_amcheck.vasp.

Orbit of Fe atoms at positions:
1 (1) [0.33333334 0.66666669 0.25        ]
2 (2) [0.66666663 0.33333331 0.75        ]
Type spin (u, U, d, D, n, N, nn or NN) for each of them (space
separated):
u d

Orbit of O atoms at positions:
3 (1) [0. 0. 0.]
4 (2) [0.  0.  0.5]
Type spin (u, U, d, D, n, N, nn or NN) for each of them (space
separated):
n n
Group of non-magnetic atoms (O): skipping.

Altermagnet? False
===========================================================
Processing: MnTe.vasp
-----------------------------------------------------------
Spacegroup: P6_3/mmc (194)
Writing the used structure to auxiliary file:
check MnTe.vasp_amcheck.vasp.

Orbit of Mn atoms at positions:
1 (1) [0. 0. 0.]
2 (2) [0.  0.  0.5]
Type spin (u, U, d, D, n, N, nn or NN) for each of them (space
separated):
u d

Orbit of Te atoms at positions:
3 (1) [0.33333334 0.66666669 0.25        ]
4 (2) [0.66666663 0.33333331 0.75        ]
Type spin (u, U, d, D, n, N, nn or NN) for each of them (space
separated):
nn
Group of non-magnetic atoms (Te): skipping.

Altermagnet? True
```

Listing 2: Interactive session example: FeO is antiferromagnetic and MnTe is alter-magnetic (as illustrated in Fig. 1). The user's input is highlighted in blue.

```python
import numpy as np
from amcheck import is_altermagnet

symmetry_operations = [(np.array([[-1,  0,   0],
                                  [ 0, -1,   0],
                                  [ 0,  0,  -1]],
                                 dtype=int),
                        np.array([0.0, 0.0, 0.0])),
                       # for compactness reasons,
                       # other symmetry operations are omitted
                       # from this example
                       ]

# positions of atoms in NiAs structure: ["Ni", "Ni", "As", "As"]
positions = np.array([[0.00, 0.00, 0.00],
                      [0.00, 0.00, 0.50],
                      [1/3., 2/3., 0.25],
                      [2/3., 1/3., 0.75]])

equiv_atoms  = [0, 0, 1, 1]

# high-pressure FeO: Fe at As positions, O at Ni positions => afm
chem_symbols = ["O", "O", "Fe", "Fe"]
spins = ["n", "n", "u", "d"]
print(is_altermagnet(symmetry_operations, positions, equiv_atoms,
                     chem_symbols, spins))

# MnTe: Mn at Ni positions, Te at As positions => am
chem_symbols = ["Mn", "Mn", "Te", "Te"]
spins = ["u", "d", "n", "n"]
print(is_altermagnet(symmetry_operations, positions, equiv_atoms,
                     chem_symbols, spins))
```

Listing 3: A python code snippet that illustrates the usage of amcheck as a library. To keep the code snippet compact and since the aim is only to illustrate the interface, only one symmetry operation is listed in this example. In order to be executed correctly, the remaining symmetry operations must be provided by the user.

```
$ amcheck --ahc RuO2.vasp
============================================================
Processing: RuO2.vasp
------------------------------------------------------------
List of atoms:
Ru [0. 0. 0.]
Ru [0.5 0.5 0.5]
O [0.30557999 0.30557999 0.        ]
O [0.19442001 0.80558002 0.5       ]
O [0.80558002 0.19442001 0.5       ]
O [0.69441998 0.69441998 0.        ]

Type magnetic moments for each atom
 ('mx my mz' or empty line for non-magnetic atom):
 1  1  0
-1 -1  0
 0  0  0
 0  0  0
 0  0  0
 0  0  0

Assigned magnetic moments:
[[1.0, 1.0, 0.0], [-1.0, -1.0, 0.0], [0, 0, 0], [0, 0, 0],
[0, 0, 0], [0, 0, 0]]

Magnetic Space Group: {'uni_number': 584,
'litvin_number': 550, 'bns_number': '65.486',
'og_number': '65.6.550', 'number': 65, 'type': 3}

Conductivity tensor:
[[ 'xx'  'xy' '-yz']
 [ 'xy'  'xx'  'yz']
 [ 'yz' '-yz'  'zz']]

The antisymmetric part of the conductivity tensor
 (Anomalous Hall Effect):
[['0'   '0' '-yz']
 ['0'   '0'  'yz']
 ['yz' '-yz' '0']]

Hall vector:
['-yz', '-yz', '0']
```

Listing 4: An example of the determination of symmetry-allowed form of the conductivity tensor for $RuO_2$ with Neel vector along [110] direction (hypothetical). The antisymmetric part of the conductivity tensor represents the form of Anomalous Hall coefficient. The user's input is highlighted in blue.

[3] L. Šmejkal, J. Sinova and T. Jungwirth, *Emerging Research Landscape of Altermagnetism*, Phys. Rev. X **12**, 040501 (2022), doi:10.1103/PhysRevX.12.040501.

[4] L. Šmejkal, R. Gonzalez-Hernandez, T. Jungwirth and J. Sinova, *Crystal time-reversal symmetry breaking and spontaneous Hall effect in collinear antiferromagnets*, Science Advances **6**, eaaz8809 (2020), doi:10.1126/sciadv.aaz8809.

[5] I. I. Mazin, K. Koepernik, M. D. Johannes, R. González-Hernández and L. Šmejkal, *Prediction of unconventional magnetism in doped FeSb2*, Proceedings of the National Academy of Sciences **118**(42) (2021), doi:10.1073/pnas.2108924118, Publisher: National Academy of Sciences Section: Physical Sciences.

[6] L. Šmejkal, A. H. MacDonald, J. Sinova, S. Nakatsuji and T. Jungwirth, *Anomalous Hall antiferromagnets*, Nature Reviews Materials **7**(6), 482 (2022), doi:10.1038/s41578-022-00430-3, Number: 6 Publisher: Nature Publishing Group.

[7] K. Samanta, M. Ležaić, M. Merte, F. Freimuth, S. Blügel and Y. Mokrousov, *Crystal Hall and crystal magneto-optical effect in thin films of SrRuO$_3$*, Journal of Applied Physics **127**(21), 213904 (2020), doi:10.1063/5.0005017.

[8] W. Feng, G. Y. Guo, J. Zhou, Y. Yao and Q. Niu, *Large magneto-optical Kerr effect in noncollinear antiferromagnets Mn3X (X=Rh, Ir, Pt)*, Physical Review B **92**(14), 144426 (2015), doi:10.1103/PhysRevB.92.144426, 1509.02865.

[9] K.-H. Ahn, A. Hariki, K.-W. Lee and J. Kuneš, *Antiferromagnetism in RuO2 as d-wave Pomeranchuk instability*, Physical Review B **99**(18), 184432 (2019), doi:10.1103/PhysRevB.99.184432.

[10] L.-D. Yuan, Z. Wang, J.-W. Luo, E. Rashba and A. Zunger, *Giant momentum-dependent spin splitting in centrosymmetric low Z antiferromagnets*, Phys. Rev. B **102**, 014422 (2020), doi:10.1103/PhysRevB.102.014422.

[11] M. Naka, S. Hayami, H. Kusunose, Y. Yanagi, Y. Motome and H. Seo, *Spin current generation in organic antiferromagnets*, Nature Communications **10**(1), 4305 (2019), doi:10.1038/s41467-019-12229-y, 1902.02506.

[12] R. González-Hernández, L. Šmejkal, K. Výborný, Y. Yahagi, J. Sinova, T. Jungwirth and J. Železný, *Efficient Electrical Spin Splitter Based on Nonrelativistic Collinear Antiferromagnetism*, Physical Review Letters **126**(12), 127701 (2021), doi:10.1103/PhysRevLett.126.127701, 2002.07073.

[13] L. Šmejkal, A. B. Hellenes, R. González-Hernández, J. Sinova and T. Jungwirth, *Giant and tunneling magnetoresistance in unconventional collinear antiferromagnets with nonrelativistic spin-momentum coupling*, Phys. Rev. X **12**, 011028 (2022), doi:10.1103/PhysRevX.12.011028.

[14] D.-F. Shao, S.-H. Zhang, M. Li, C.-B. Eom and E. Y. Tsymbal, *Spin-neutral currents for spintronics*, Nature Communications **12**(1), 7061 (2021), doi:10.1038/s41467-021-26915-3.

[15] Y. Fang, J. Cano and S. A. A. Ghorashi, *Quantum geometry induced nonlinear transport in altermagnets*, doi:10.48550/arXiv.2310.11489 (2023), 2310.11489.

[16] R. M. Fernandes, V. S. de Carvalho, T. Birol and R. G. Pereira, *Topological transition from nodal to nodeless Zeeman splitting in altermagnets*, doi:10.48550/arXiv.2307.12380 (2023), 2307.12380.

[17] S.-B. Zhang, L.-H. Hu and T. Neupert, *Finite-momentum cooper pairing in proximitized altermagnets*, doi:10.48550/arXiv.2302.13185 (2023), 2302.13185.

[18] C. Sun, A. Brataas and J. Linder, *Andreev reflection in altermagnets*, Phys. Rev. B **108**, 054511 (2023), doi:10.1103/PhysRevB.108.054511.

[19] J. A. Ouassou, A. Brataas and J. Linder, *dc Josephson Effect in Altermagnets*, Phys. Rev. Lett. **131**, 076003 (2023), doi:10.1103/PhysRevLett.131.076003.

[20] L. Šmejkal, A. Marmodoro, K.-H. Ahn, R. González-Hernández, I. Turek, S. Mankovsky, H. Ebert, S. W. D'Souza, O. Šipr, J. Sinova and T. Jungwirth, *Chiral Magnons in Altermagnetic* $RuO_2$, Phys. Rev. Lett. **131**, 256703 (2023), doi:10.1103/PhysRevLett.131.256703.

[21] S. Bhowal and N. A. Spaldin, *Magnetic octupoles as the order parameter for unconventional antiferromagnetism*, arXiv:2212.03756 (2022), doi:arXiv.2212.03756.

[22] I. I. Mazin, *Notes on altermagnetism and superconductivity*, doi:10.48550/arXiv.2203.05000 (2022), 2203.05000.

[23] J. Krempaský, L. Šmejkal, S. W. D'Souza, M. Hajlaoui, G. Springholz, K. Uhlířová, F. Alarab, P. C. Constantinou, V. Strokov, D. Usanov, W. R. Pudelko, R. González-Hernández *et al.*, *Altermagnetic lifting of Kramers spin degeneracy*, doi:10.48550/arXiv.2308.10681 (2023), 2308.10681.

[24] S. A. Egorov and R. A. Evarestov, *Colossal Spin Splitting in the Monolayer of the Collinear Antiferromagnet MnF2*, The Journal of Physical Chemistry Letters **12**(9), 2363 (2021), doi:10.1021/acs.jpclett.1c00282, Publisher: American Chemical Society.

[25] C. J. Bradley and A. P. Cracknell, *The mathematical theory of symmetry in solids: representation theory for point groups and space groups*, Oxford classic texts in the physical sciences. Clarendon Press, Oxford, ISBN 978-0-19-958258-7 (2010).

[26] K. Momma and F. Izumi, *VESTA3 for three-dimensional visualization of crystal, volumetric and morphology data*, Journal of Applied Crystallography **44**(6), 1272 (2011), doi:10.1107/S0021889811038970.

[27] Y. Fei and H. kwang Mao, *In Situ Determination of the NiAs Phase of FeO at High Pressure and Temperature*, Science **266**(5191), 1678 (1994), doi:10.1126/science.266.5191.1678.

[28] Kunitomi, Nobuhiko, Hamaguchi, Yoshikazu and Anzai, Shuichiro, *Neutron diffraction study on manganese telluride*, J. Phys. France **25**(5), 568 (1964), doi:10.1051/jphys:01964002505056800.

[29] R. Toft-Petersen, N. H. Andersen, H. Li, J. Li, W. Tian, S. L. Bud'ko, T. B. S. Jensen, C. Niedermayer, M. Laver, O. Zaharko, J. W. Lynn and D. Vaknin, *Magnetic phase diagram of magnetoelectric LiMnPO$_4$*, Phys. Rev. B **85**, 224415 (2012), doi:10.1103/PhysRevB.85.224415.

[30] S. Gnewuch and E. E. Rodriguez, *Distinguishing the Intrinsic Antiferromagnetism in Polycrystalline LiCoPO4 and LiMnPO4 Olivines*, Inorganic Chemistry **59**(9), 5883 (2020), doi:10.1021/acs.inorgchem.9b03545, PMID: 32319759.

[31] *International tables for crystallography*, International Union of Crystallography, 1st online edn., ISBN 9780792365907, doi:10.1107/97809553602060000100 (2006).

[32] A. Togo and I. Tanaka, Spglib: *a software library for crystal symmetry search*, doi:10.48550/ARXIV.1808.01590 (2018).

[33] *spglib website*, https://spglib.github.io/spglib/index.html.

[34] *As an exceptional case, if a supercell (or non-primitive cell) has the basis vectors whose lattice breaks crystallographic point group, the crystallographic symmetry operations are searched within this broken symmetry, i.e., at most the crystallographic point group found in this case is the point group of the lattice. for example, this happens for the supercell of a conventional cubic unit cell. this may not be understandable in crystallographic sense, but is practically useful treatment for research in computational materials science.*, https://spglib.github.io/spglib/api.html?highlight=supercell#spg-get-symmetry.

[35] G. L. W. Hart and R. W. Forcade, *Algorithm for generating derivative structures*, Phys. Rev. B **77**, 224115 (2008), doi:10.1103/PhysRevB.77.224115.

[36] R. Jaeschke-Ubiergo, V. K. Bharadwaj, T. Jungwirth, L. Šmejkal and J. Sinova, *Supercell Altermagnets* (2023), 2308.16662.

[37] L. M. Corliss, N. Elliott and J. M. Hastings, *Antiferromagnetic Structures of MnS2, MnSe2, and MnTe2*, Journal of Applied Physics **29**(3), 391 (2004), doi:10.1063/1.1723149.

[38] A. H. Larsen, J. J. Mortensen, J. Blomqvist, I. E. Castelli, R. Christensen, M. Dułak, J. Friis, M. N. Groves, B. Hammer, C. Hargus, E. D. Hermes, P. C. Jennings *et al.*, *The atomic simulation environment—a Python library for working with atoms*, Journal of Physics: Condensed Matter **29**(27), 273002 (2017), doi:10.1088/1361-648X/aa680e.

[39] *Atomic Simulation Environment*, https://wiki.fysik.dtu.dk/ase/.

[40] H. J. Bernstein, J. C. Bollinger, I. D. Brown, S. Gražulis, J. R. Hester, B. McMahon, N. Spadaccini, J. D. Westbrook and S. P. Westrip, *Specification of the Crystallographic Information File format, version 2.0*, Journal of Applied Crystallography **49**(1), 277 (2016), doi:10.1107/S1600576715021871.

[41] *Description of* POSCAR *structure file format*, https://www.vasp.at/wiki/index.php/POSCAR.

[42] S. V. Gallego, J. Etxebarria, L. Elcoro, E. S. Tasci and J. M. Perez-Mato, *Automatic calculation of symmetry-adapted tensors in magnetic and non-magnetic materials: a new tool of the Bilbao Crystallographic Server*, Acta Crystallographica Section A: Foundations and Advances **75**(3), 438 (2019), doi:10.1107/S2053273319001748, Publisher: International Union of Crystallography.