# Peer review of "A tool to check whether a symmetry-compensated collinear magnetic material is antiferro- or altermagnetic"

_SciPost Physics Codebases, doi:SciPost Phys. Codebases 30 (2024) , SciPost Phys. Codebases 30-r1.0 (2024)_

## Round 1 · Referee Report · Anonymous (Referee 1) · 2024-1-26

Strengths

  • Provides a reasonably accessible code which can separate AFM and AM materials, provided that a certain set of assumptions are satisfied
  • Clear explanation of fundamental symmetry properties distinguishing AFM and AM materials

Weaknesses

  • Neglects spin-orbit interactions, even weak, since spin and space coordinates are considered to be decoupled
  • In the case of multiple magnetic ions in the crystal structure, a restriction on the code is that each Wyckoff orbit of magnetic atoms has zero net magnetic moment

Report

Given the recent surge of interest in altermagnetic (AM) materials, this paper provides a timely code which can be used to distinguish antiferromagnet (AFM) materials with a spin-degenerate electron band structure from AM materials which feature a momentum-dependent spin-splitting.

Overall, the paper is well written. The code only appears to be valid under a set of restrictions, such as neglecting spin-orbit interactions. Despite this, I think the paper will be useful to specialists in the field. Before I can recommend it for publication, I would ask the authors to address two minor issues listed below.

Requested changes

(1) At the end of section 2.1, the authors write "Thus, to check if the material
is AM, one needs to confirm the absence of these symmetry operations while confirming that
there is some symmetry followed by a spin-flip that relates up and down sublattices". Everything in this section is clear to me up to this last sentence. It is in particular the part "... ome symmetry followed by a spin-flip that relates up and down sublattices" which is unclear to me. Namely, in a conventional AFM material, the system is invariant under an inversion + a spin-flip. Isn't this precisely the type of operation the authors refer to in the last part of the statement? It looks like the authors are writing that one needs to confirm that e.g. inversion + spin-flip does not leave the system invariant, and then immediately after they write that there should exist an operation (such as inversion) + spin-flip which leaves the system invariant. Please clarify.

(2) In the beginning of section 2.2 and the figure caption of Figure 1, the authors refer to an NiAs structure. Initially, this confused me a bit since the authors in the main write that As atoms are blue in fig 1a whereas Ni atoms are red. But in the actual figure, it is stated that fig 1a shows FeO. It would be helpful for the reader if the authors clarify precisely what is meant by an NiAs structure (namely the geometrical structure) whereas the materials they actually consider are different from Ni and As.

---

## Round 2 · Author Response

>> Weaknesses
>> - Neglects spin-orbit interactions, even weak, since spin and space coordinates are considered to be decoupled
This is simply a misunderstanding. The purpose of the code is to determine whether the material is altermagnetic or not, and the answer to this question, by the very definition of altermagnetism, requires analysis of the nonrelativistic symmetries, i.e., without spin-orbit interaction (SOI). After having established that the material in question is AM, one can ask the next question: what kind of anomalous transport is allowed by symmetry in this AM, if any? In answering this question, we do not neglect SOI, and, in fact, it is instrumental for the program switch “-ahc”. So, basically, the program can be used in two regimes: one is an analysis of spin group symmetry, where SOI is not present by construction, and magnetic group symmetry, where it plays a key role. We have made it explicit in the revised version.
>> - In the case of multiple magnetic ions in the crystal structure, a restriction on the code is that each Wyckoff orbit of magnetic atoms has zero net magnetic moment
This is also a misunderstanding. If a Wyckoff orbit has a nonzero net moment, this means that the material in question is ferrimagnetic and not altermagnetic, and no further analysis is needed. Again, we now state it explicitly.
>> The code only appears to be valid under a set of restrictions, such as neglecting spin-orbit interactions.
As mentioned above, this is a misunderstanding. The code is completely general and does not have any restrictions.
>> (1) At the end of section 2.1, the authors write "Thus, to check if the materialis AM, one needs to confirm the absence of these symmetry operations while confirming that there is some symmetry followed by a spin-flip that relates up and down sublattices". Everything in this section is clear to me up to this last sentence. It is in particular the part "... ome symmetry followed by a spin-flip that relates up and down sublattices" which is unclear to me. Namely, in a conventional AFM material, the system is invariant under an inversion + a spin-flip. Isn't this precisely the type of operation the authors refer to in the last part of the statement? It looks like the authors are writing that one needs to confirm that e.g. inversion + spin-flip does not leave the system invariant, and then immediately after they write that there should exist an operation (such as inversion) + spin-flip which leaves the system invariant. Please clarify.
We see how this sentence could be confusing. We have rewritten this entire paragraph to make it clearer. It now reads:
As we see, if the net magnetization is strictly zero, it may be not dictated by symmetry, which is the case of a Luttinger-compensated ferrimagnet, or result from one or more crystal symmetry operation mapping one spin sublattice upon the other. This symmetry operation may be (a) a lattice translation, (b) spatial inversion or (c) another operation that is neither translation nor inversion. If all symmetry operations performing such matting belong to the group (c), the material is altermagnetic and its bands are spin-split at general k-points. Thus, to check if the material is AM, one needs to confirm that there are no translations and no inversions that map one spin sublattice onto the other.
>> (2) In the beginning of section 2.2 and the figure caption of Figure 1, the authors refer to an NiAs structure. Initially, this confused me a bit since the authors in the main write that As atoms are blue in fig 1a whereas Ni atoms are red. But in the actual figure, it is stated that fig 1a shows FeO. It would be helpful for the reader if the authors clarify precisely what is meant by an NiAs structure (namely the geometrical structure) whereas the materials they actually consider are different from Ni and As
Yes, by the NiAs structure we merely mean the structure prototype, as it is common in crystallography, and different magnetic phases are discussed depending on what material derived from this prototype is under consideration. Since this appears to be confusing, we now introduce a figure with the NiAs structure prototype, which is discussed in the first part of the relevant paragraph.

---

## Editorial Decision

published